# MIMIC-Sepsis: A Curated Benchmark for Modeling and Learning from Sepsis Trajectories in the ICU

Yong Huang
*Department of Computer Science*
*University of California, Irvine*
Irvine, California
yongh7@uci.edu

Zhongqi Yang
*Department of Computer Science*
*University of California, Irvine*
Irvine, California
zhongqy4@uci.edu

Amir Rahmani
*Department of Computer Science*
*University of California, Irvine*
Irvine, California
a.rahmani@uci.edu

*Abstract*—Sepsis is a leading cause of mortality in intensive care units (ICUs), yet existing research often relies on outdated datasets, non-reproducible preprocessing pipelines, and limited coverage of clinical interventions. We introduce MIMIC-Sepsis, a curated cohort and benchmark framework derived from the MIMIC-IV database, designed to support reproducible modeling of sepsis trajectories. Our cohort includes 35,239 ICU patients with time-aligned clinical variables and standardized treatment data, including vasopressors, fluids, mechanical ventilation and antibiotics. We describe a transparent preprocessing pipeline—based on Sepsis-3 criteria, structured imputation strategies, and treatment inclusion—and release it alongside benchmark tasks focused on early mortality prediction, length-of-stay estimation, and shock onset classification. Empirical results demonstrate that incorporating treatment variables substantially improves model performance, particularly for Transformer-based architectures. MIMIC-Sepsis serves as a robust platform for evaluating predictive and sequential models in critical care research.

*Index Terms*—Sepsis, benchmark, MIMIC-IV, machine learning, clinical data, intensive care unit, prediction models

## I. INTRODUCTION

Sepsis is a life-threatening condition caused by the body's extreme response to an infection that can lead to organ failure and even death. It is one of the leading causes of death in the world. According to the World Health Organization, there are an estimated 48.9 million cases and 11 million sepsis-related deaths worldwide, representing 20% of all global deaths [1]. In addition, the cost of sepsis is staggering; the average hospital-wide cost of sepsis was estimated to be more than US$32,000 per patient in high-income countries [1]. The onset of sepsis is often acute and can be difficult to detect, which may result in delayed treatment and, consequently, irreversible organ damage. As such, early diagnosis and timely interventions are crucial for improving patient survival rates. Recent studies have shown that early administration of vaso-pressors is associated with increased survival rates in patients with septic shock, while delayed administration of antibiotics after sepsis identification significantly increases in-hospital mortality rates [2], [3].

Over the past few decades, several large-scale EHR datasets collected from intensive care units (ICU) have been made publicly available for research purposes, including MIMIC-III and eICU [4], [5]. These datasets encompass diverse patient populations and clinical conditions. One important feature of these datasets is the inclusion of extensive clinical variables, including physiological measurements, laboratory test results, medication administration records, and detailed documentation of clinical interventions, which builds a solid foundation for developing and evaluating novel computational approaches for sepsis research.

Researchers have applied state-of-the-art machine learning models to these datasets, yielding significant advances in fore-casting sepsis progression and patient outcomes. Deep learning methodologies have been employed to predict in-hospital mortality among septic patients with promising results [6]. Furthermore, reinforcement learning algorithms have been shown to optimize therapeutic interventions—specifically the dosage and timing of vasopressors and intravenous fluid ad-ministration—potentially surpassing human clinical decision-making in improving patient survival rates [7].

Despite these methodological innovations and growing in-terest in leveraging large-scale ICU datasets for sepsis re-search, significant practical challenges persist. First, many existing studies rely on the outdated and relatively smaller MIMIC-III dataset, which may no longer reflect current clinical practices. The newer MIMIC-IV dataset [8] offers a more recent and comprehensive resource, yet efforts to curate standardized sepsis cohorts from MIMIC-IV remain limited. Second, data preprocessing procedures are often in-consistent and poorly documented. Preparing these complex clinical datasets for modeling requires extensive extraction, harmonization, and cleaning. Most prior sepsis studies lack transparent documentation of these steps, which significantly impedes reproducibility and validation. Finally, clinical in-terventions are frequently overlooked in analyses of sepsis trajectories. However, treatments such as vasopressors, flu-ids, and mechanical ventilation can dramatically influence physiological variables—for example, blood pressure may rise after vasopressor administration, or oxygen saturation may improve following oxygen therapy. Failure to account for these interventions can lead to biased conclusions about disease progression and treatment efficacy.

In this study, we introduce **MIMIC-Sepsis**, a curated co-hort and benchmark framework derived from the MIMIC-IV database. The cohort comprises 35,239 ICU stays that satisfy

Sepsis-3 criteria, with curated time-aligned features including vital signs, laboratory results, and treatment interventions such as vasopressors, intravenous fluids, and antibiotics. We standardize dosage units, apply multi-level imputation strategies, and transform event-based clinical data into structured longitudinal format to facilitate modeling. In addition to releasing the cohort and processing pipeline, we define a set of benchmark tasks—including mortality prediction, length-of-stay estimation, and shock classification—to enable reproducible and extensible evaluation of predictive and sequential models in sepsis care.

Our contributions are twofold: (1) we provide a transparent, reproducible cohort construction pipeline with harmonized clinical variables, and (2) we introduce an evaluation framework that supports time-aware modeling of treatment dynamics and outcomes. Through empirical experiments, we find that integrating temporal sequences of clinical data with treatment information improves model performance, particularly for Transformer-based architectures. MIMIC-Sepsis aims to serve as a public benchmark for advancing machine learning applications in sepsis and critical care.

The code and data processing pipeline are publicly available at: https://github.com/yongh7/MIMIC-sepsis.

## II. Related Work

Early detection of sepsis or septic shock onset remains a critical challenge in improving patient outcomes. Calvert et al. [9] were among the first to address this challenge by developing a time series-based regression model using the MIMIC-II database to predict sepsis 3 hours prior to onset for patients in the Medical Intensive Care Unit (MICU). Subsequent research has expanded upon this work with more sophisticated methodologies and datasets. For instance, Fagerstrom et al. [10] employed a Cox proportional hazards model on MIMIC-III data to predict septic shock onset, while Deshon et al. [11] applied survival analysis techniques to predict sepsis onset using a proprietary dataset. Additionally, Goh et al. [12] explored an alternative approach utilizing unstructured clinical text from MIMIC-III for sepsis prediction.

Predicting outcomes in sepsis represents another active research topic. Notable contributions include Hou et al. [13], who implemented tree-based models to predict 30-day mortality, and Yong et al. [6] , who developed deep learning architectures to forecast in-hospital mortality, both utilizing laboratory and vital sign measurements from MIMIC-III. Furthermore, Boussina et al. [14] conducted a before-and-after quasi-experimental study evaluating the clinical impact of a deep learning model for early sepsis prediction within the UC San Diego Health system.

Treatment optimization constitutes a distinct domain within computational sepsis research. Raghu et al. [7] applied reinforcement learning algorithms to optimize therapeutic interventions for sepsis patients, specifically addressing the dosage and timing of vasopressors and intravenous fluid administration. Huang et al. [15] refined this approach by implementing more granular control of treatment dosing strategies.

Additionally, Choudhary et al. [16] introduced ICU-sepsis, an environment built upon the MIMIC-III dataset that provides standardized benchmarks for evaluating reinforcement learning algorithms in sepsis treatment, focusing on the same two interventions (vasopressors and intravenous fluids).

While these studies represent significant contributions to sepsis research, they exhibit several limitations: reliance on outdated datasets (MIMIC-II and MIMIC-III), employment of ad-hoc data curation processes lacking standardization and reproducibility, narrow focus on isolated aspects of sepsis care, and insufficient inclusion of comprehensive clinical interventions. Notably, antibiotic administration—an important aspect of sepsis treatment—is frequently overlooked in analyses despite its critical importance in clinical practice.

## III. Dataset/Benchmark Design

### A. Benchmark Design Principles

Our dataset and benchmark framework are built around three core principles. First, we preserve the temporal structure of clinical trajectories by aligning all events relative to the suspected sepsis onset. This supports time-sensitive analyses, such as early warning prediction and treatment effect modeling. Second, we explicitly incorporate clinical interventions—including vasopressors, fluids, antibiotics, and mechanical ventilation—acknowledging their influence on patient physiology and outcomes. Third, we ensure complete reproducibility by releasing our data preprocessing pipeline and providing transparent documentation of each step, including cohort inclusion criteria, temporal alignment, imputation methods, and variable definitions.

### B. Cohort Selection and Data Processing

Our cohort construction process draws on methodologies from prior sepsis studies on MIMIC-III by Komorowski et al. [17] and Killian et al. [18]. We extract three categories of data: (1) static demographics such as age, sex, and Charlson comorbidity index; (2) longitudinal clinical measurements including vitals, laboratory tests, microbiology cultures, and urine output; and (3) clinical interventions relevant to sepsis management.

To identify suspected infection, we follow the Sepsis-3 definition [19], using either antibiotic administration records or positive microbiological cultures as potential infection triggers. Once the presumed infection time is determined, we define a fixed observational window extending from 24 hours before to 72 hours after infection onset. This window is designed to capture both early detection signals and the acute progression phase of sepsis.

Clinical measurements within this window are standardized to consistent units, and implausible outliers are removed based on clinical thresholds. Data are resampled into fixed 4-hour intervals. When multiple values exist within an interval, we compute the mean. For missing values, we apply forward fill if a prior value exists within a variable-specific validity window (e.g., longer for stable measures like weight). Remaining missingness is addressed through a tiered imputation strategy

inspired by Komorowski et al. [17]. Specifically, Linear interpolation is used for variables with low missingness ($< 5\%$) to preserve local temporal trends without introducing complex assumptions, while K-nearest neighbors (KNN) imputation is applied for moderate missingness to exploit correlations across similar patient profiles in the multivariate feature space. Variables with more than $80\%$ missingness are excluded due to insufficient data support and the high risk of introducing bias through imputation. Certain variables are estimated using clinical rules—for instance, $FiO_2$ is derived from oxygen flow rate and device type, and GCS is inferred from RASS scores [20]. From the cleaned data, we compute derived scores such as SOFA and SIRS to assess organ dysfunction and systemic inflammation.

We extract four types of interventions central to sepsis care: (1) mechanical ventilation (mode and parameters), (2) antibiotics (timing and number of unique agents), (3) fluid resuscitation (standardized to NaCl 0.9% equivalent volume), and (4) vasopressors (converted to norepinephrine-equivalent dosage). For each 4-hour interval, we compute cumulative vasopressor dose and fluid volume. Including these treatment variables allows us to capture not just patient status, but also care dynamics.

Sepsis onset is identified using Sepsis-3 criteria: the earliest timepoint where a patient's SOFA score increases by two or more points from baseline in the presence of infection. Septic shock is defined using three conditions: (1) administration of at least 2000 mL of fluids in the prior 12 hours, (2) MAP $<$ 65 mmHg despite fluids, and (3) vasopressor requirement with lactate $>$ 2 mmol/L [19].

Finally, we exclude patients under 18 years of age, non-sepsis patients, those with implausible fluid input/output values, and individuals who died shortly after ICU admission—potentially indicating withdrawal of care.

## IV. Dataset/Benchmark Description

Now we present the characteristics of the curated sepsis cohort and proposed benchmark.

### A. Data statistics

This section presents the statistics of the curated sepsis cohort and proposed benchmark. The dataset comprises 35,239 patients with clinical variables tracked over time. Table I summarizes the key demographic and clinical characteristics of the cohort.

The cohort represents a diverse population of sepsis patients with varying degrees of disease severity. The majority of patients are middle-aged or elderly, with a gender distribution slightly skewed toward males. The median Charlson comorbidity index of 5.0 suggests a considerable burden of comorbid conditions within this population. Clinical parameters show considerable variability. Figure 2 illustrates the distribution of key physiological measurements across various organ systems among the selected cohort, along with their inter-parameter correlations. The majority of clinical parameters demonstrate

**TABLE I.** Cohort Characteristics.

| Characteristic | Value |
| --- | --- |
| **Demographics** | |
| Total patients | 35,239 |
| Age (mean ± std) | 65.4 ± 16.3 |
| Gender (% female) | 44.5% |
| Charlson Index (median [IQR]) | 5.0 [3.0-7.0] |
| **Age Distribution (%)** | |
| 18-40 | 8.5% |
| 41-65 | 38.1% |
| 66-80 | 33.9% |
| $> 80$ | 19.4% |
| **BMI Distribution (%)** | |
| Underweight ($< 18.5$) | 7.2% |
| Normal (18.5-24.9) | 29.0% |
| Overweight (25-29.9) | 26.5% |
| Obese ($\geq 30$) | 37.3% |
| **Clinical Outcomes** | |
| Hospital mortality (%) | 14.5% |
| 90-day mortality (%) | 25.1% |
| Length of stay (days) | 5.1 ± 7.1 |
| Readmission rate (%) | 7.3% |
| Septic shock (%) | 12.4% |
| **Disease Severity** | |
| SOFA score (mean ± std) | 5.5 ± 2.8 |
| SIRS score (mean ± std) | 1.5 ± 1.0 |
| **Interventions** | |
| Mechanical ventilation (%) | 35.1% |
| Vasopressor use (%) | 16.9% |
| Antibiotics given (%) | 66.3% |
| Patients receiving all 3 interventions (%) | 23.3% |
| **Fluid Management** | |
| Mean fluid balance (mL) | -931.2 ± 8142.5 |
| Fluid rate per 4h (mL) | 458.9 ± 645.2 |
| Cumulative fluid at 24h (mL) | 3017.3 ± 5566.5 |
| Patients with negative fluid balance (%) | 56.9% |

approximately normal distributions, with the exception of the Glasgow Coma Scale (GCS) score.

Most patients in the study received antibiotics, totaling 66.3%. And a smaller percentage required mechanical ventilation (35.1%) and vasopressors (16.9%). In Figure 3 we observed that patients receiving antibiotics or vasopressors exhibit higher mortality rates, and interestingly, those who received these treatments tend to have worse clinical outcomes. However, these observations likely reflect confounding by indication, as sicker patients typically receive treatments earlier.

### B. Benchmark Tasks

We define four predictive tasks designed to reflect real-world challenges in sepsis care—from early risk stratification to dynamic treatment planning. These tasks are grouped into two categories based on their temporal structure:

1) **Static Prediction from Early Observations**: For in-hospital mortality (IHM) and length of stay (LOS), models use only the first 6 hours of the sepsis trajectory. This simulates early clinical decision-making scenarios where rapid risk assessment is needed to guide treatment and resource allocation.

2) **Dynamic Prediction with Rolling Windows**: For vasopressor requirement (VR) and septic shock (SS), we use a rolling window approach. At each time step, the model observes the previous 6 hours of data to predict whether

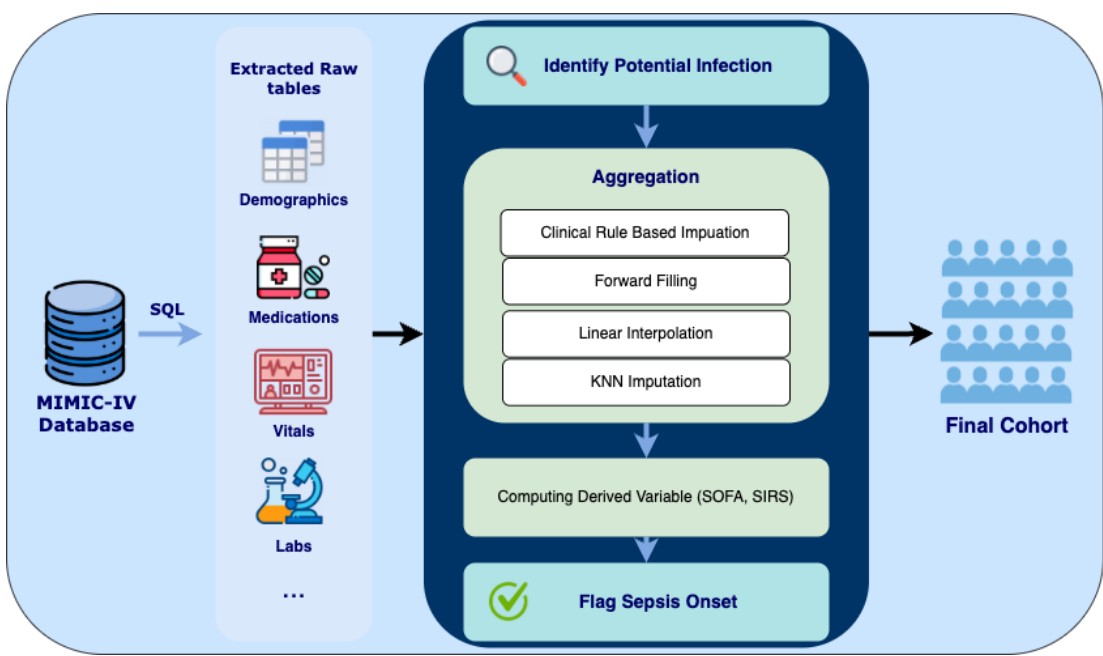

**Fig. 1.** Cohort selection and data processing workflow. This figure illustrates the high-level approach used to extract and process sepsis-related data from MIMIC-IV.

the event will occur within the next 24 hours. This mirrors real-time monitoring environments that require continuous, updated assessments of patient status.

Table II summarizes the prediction tasks, modeling setup, and evaluation metrics. IHM, VR and SS are framed as binary classification problems, and LOS as regression. We standardize metrics across tasks to enable consistent and fair comparisons between models.

**TABLE II.** Benchmark Tasks. IHM: In-hospital Mortality, LOS: Length of Stay, VR: Vasopressor Requirement, SS: Septic Shock.

| Task | Type | Approach | Metrics |
|------|------|----------|---------|
| IHM | Binary Class. | Static | AUROC, AUPRC |
| LOS | Regression | Static | MAE, RMSE |
| VR | Binary Class. | Dynamic | AUROC, AUPRC |
| SS | Binary Class. | Dynamic | AUROC, AUPRC |

These standardized tasks are intended to facilitate robust benchmarking and reproducibility, supporting the development and evaluation of new machine learning models in sepsis and critical care research.

## V. EXPERIMENTS AND RESULTS

We evaluate model performance on the benchmark tasks using a range of machine learning algorithms, from linear baselines to deep learning architectures including LSTMs and Transformers [21], [22]. The dataset is randomly split into 80% for training and 20% for testing.

For regression tasks (e.g., length of stay prediction), we report Mean Absolute Error (MAE) and Root Mean Squared Error (RMSE). For classification tasks (e.g., mortality and septic shock prediction), we report the Area Under the Receiver Operating Characteristic curve (AUROC) and Area Under the Precision-Recall Curve (AUPRC). Compared to metrics such as accuracy or precision, AUROC and AUPRC provide more robust evaluations in the presence of class imbalance. For example, a naive model that always predicts the majority class may achieve high accuracy but only 0.5 AUROC, indicating no true discrimination ability.

We test three classes of models:

- **Linear Model:** Temporal features within the observation window are flattened and used as input to a single-layer perceptron.
- **LSTM:** A standard two-layer LSTM network is used, with a dropout rate of 0.1 to prevent overfitting.
- **Transformer:** A transformer encoder with 8 attention heads and 2 layers is implemented, also using a dropout rate of 0.1 to ensure fair comparison with the LSTM model.

A key contribution of our benchmark is the incorporation of clinical treatment variables. To quantify their utility, we compare model performance with and without treatment-related features (e.g., vasopressor dosage, fluid intake, antibiotic exposure). To our knowledge, this is the first sepsis benchmark to systematically evaluate the impact of incorporating treatment variables into predictive modeling.

### A. Results and Analysis

Table III presents the performance of the models on each benchmark task with treatment variables included. Table IV shows the same tasks evaluated without treatment variables,

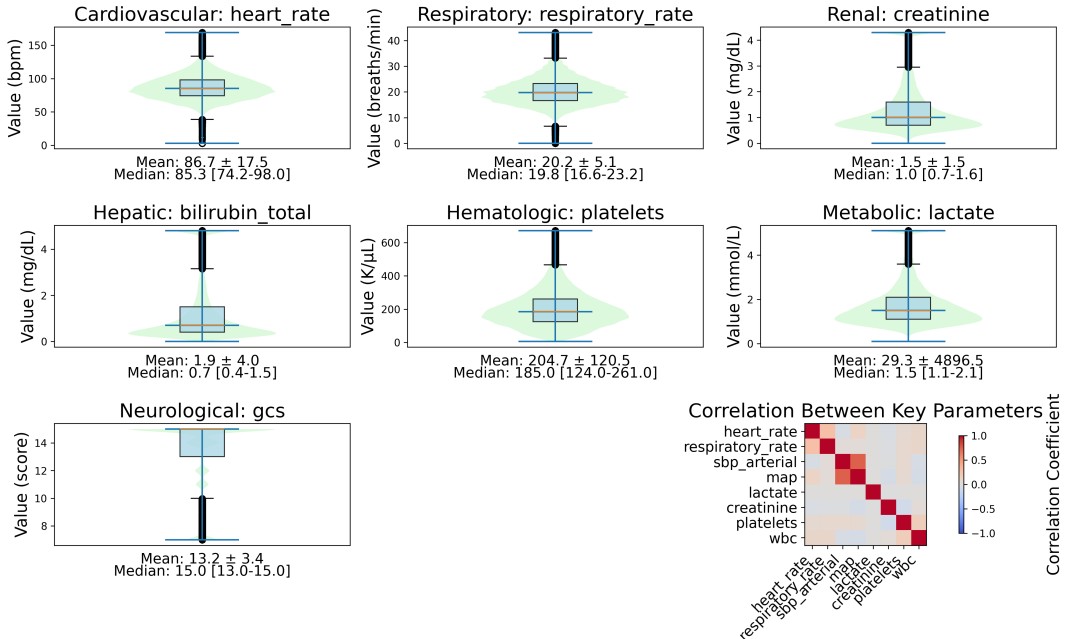

**Fig. 2.** Key Clinical Parameters Across Organ Systems. Each panel shows the distribution of a key parameter using boxplots and violin plots. The correlation heatmap (bottom right) displays relationships between parameters.

enabling a direct comparison of the effect of including clinical interventions.

**TABLE III.** Model Performance on Benchmark Tasks (With Treatment Variables). IHM: In-hospital Mortality, LOS: Length of Stay, SS: Septic Shock, VR: Vasopressor Requirement. Best performance for each metric is bolded.

| Task | Metric | Linear | LSTM | Transformer |
|------|--------|--------|------|-------------|
| IHM | AUROC ↑ | 0.845 | 0.838 | **0.863** |
| IHM | AUPRC ↑ | 0.512 | 0.507 | **0.550** |
| LOS | RMSE ↓ | 13.22 | 5.23 | **5.12** |
| LOS | MAE ↓ | 3.10 | 2.85 | **2.81** |
| SS | AUROC ↑ | 0.881 | 0.885 | **0.925** |
| SS | AUPRC ↑ | 0.497 | 0.580 | **0.705** |
| VR | AUROC ↑ | 0.924 | 0.911 | **0.927** |
| VR | AUPRC ↑ | 0.892 | 0.870 | **0.903** |

The results show that deep learning models, particularly the Transformer architecture, consistently outperform the linear baseline across all benchmark tasks. The Transformer achieves the highest AUROC and AUPRC scores in mortality and septic shock prediction, and demonstrates the lowest RMSE and MAE in the length of stay task. Similarly, the vasopressor requirement task benefits from deep learning models, with the Transformer again showing superior performance.

A key finding is the substantial improvement in dynamic prediction tasks when treatment variables are included. For example, in the vasopressor requirement task, the LSTM

**TABLE IV.** Model Performance on Benchmark Tasks (Without Treatment Variables). IHM: In-hospital Mortality, LOS: Length of Stay, SS: Septic Shock, VR: Vasopressor Requirement. Best performance for each metric is bolded.

| Task | Metric | Linear | LSTM | Transformer |
|------|--------|--------|------|-------------|
| IHM | AUROC ↑ | 0.844 | 0.825 | **0.864** |
| IHM | AUPRC ↑ | 0.512 | 0.487 | **0.560** |
| LOS | RMSE ↓ | 13.81 | 5.31 | **5.18** |
| LOS | MAE ↓ | 3.14 | 2.86 | **2.70** |
| SS | AUROC ↑ | 0.876 | 0.879 | **0.919** |
| SS | AUPRC ↑ | 0.489 | 0.501 | **0.672** |
| VR | AUROC ↑ | 0.823 | 0.779 | **0.810** |
| VR | AUPRC ↑ | 0.697 | 0.635 | **0.687** |

model sees an absolute increase of 0.112 in AUROC and 0.235 in AUPRC when treatment features are added.

For static prediction tasks like in-hospital mortality and length of stay, the impact of treatment variables is less pronounced. We hypothesize that this is due to the limited observation window (first 6 hours), during which treatment effects may not yet be fully manifested.

We also conducted ablation experiments varying the prediction horizon for dynamic tasks and observed that model performance was relatively robust to horizon length, suggesting that the rolling window design generalizes well to different clinical settings.

While our benchmark focuses on four core tasks, the

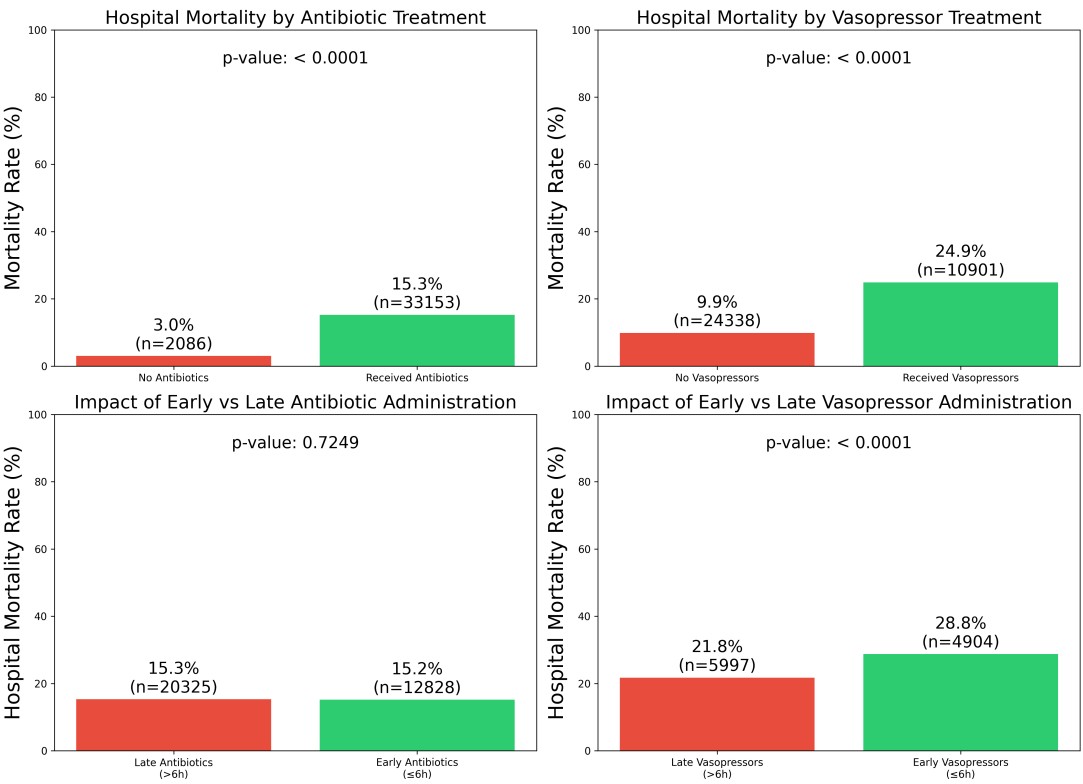

**Fig. 3.** Association between treatment interventions and hospital mortality in sepsis patients. The upper panels compare mortality rates between patients who did and did not receive antibiotics (left) and vasopressors (right). The lower panels analyze the impact of early ($\leq$ 6h) versus late ($>$ 6h) administration timing. Statistical significance was assessed using chi-square tests, with p-values displayed for each comparison.

framework is extensible. For example, additional targets such as the need for mechanical ventilation or readmission risk can be integrated based on specific research objectives. We also release baseline implementations to serve as a foundation for future work and facilitate model development for clinical decision support systems aimed at early risk identification and treatment optimization.

## VI. DISCUSSION

We introduced MIMIC-Sepsis, a curated dataset and benchmark framework designed to advance machine learning research in sepsis care. Our work contributes a reproducible cohort construction pipeline with harmonized clinical variables and a suite of benchmark tasks that reflect clinically meaningful challenges. By aligning data relative to sepsis onset and incorporating treatment interventions such as vasopressors, antibiotics, fluids, and ventilation, our benchmark enables time-aware modeling of patient trajectories and treatment outcomes.

Our findings highlight the importance of including treatment variables in predictive modeling. While their impact was more pronounced in dynamic tasks (e.g., vasopressor requirement), their limited effect in early static prediction tasks suggests

future work could explore better representations or incorporate interaction effects between physiology and interventions. The results also affirm that Transformer-based architectures benefit from this enriched temporal structure and intervention information.

Several limitations should be acknowledged. First, the dataset is derived from a single institution (Beth Israel Deaconess Medical Center), which may affect generalizability to other settings. Second, as with all retrospective EHR data, inconsistencies in documentation and potential biases in care delivery may influence the extracted variables. Third, while we adopted the widely accepted Sepsis-3 definition, newer tools like qSOFA and advanced severity scoring systems are not yet integrated and may improve future cohort definitions.

Looking ahead, the MIMIC-Sepsis benchmark provides a strong foundation for more complex modeling tasks. One promising direction is the development of reinforcement learning algorithms to guide sequential decision-making in sepsis care, particularly for treatment personalization. Our structured and time-aligned dataset is readily suitable for this class of models.

Another important extension is the incorporation of unstruc-

tured clinical data such as discharge summaries and progress notes. Though not yet included in our benchmark tasks, our preprocessing tools provide alignment between structured and unstructured data, allowing researchers to experiment with multimodal modeling approaches. These notes contain rich contextual insights that may enhance risk stratification, trajectory forecasting, and causal inference.

Overall, we aim for MIMIC-Sepsis to serve as a standardized and extensible benchmark that supports the reproducibility, comparability, and clinical relevance of machine learning research in critical care.

## VII. Conclusion

This work introduces MIMIC-Sepsis, a publicly available benchmark designed to support machine learning research in sepsis and critical care. We contribute: (1) a transparent and reproducible pipeline for cohort construction and variable harmonization; and (2) a benchmark framework that incorporates treatment interventions and supports time-aware modeling of disease progression and outcomes.

By aligning clinical events relative to sepsis onset and including interventions such as vasopressors, fluids, antibiotics, and ventilation, MIMIC-Sepsis enables the study of treatment dynamics often overlooked in prior sepsis research. Our experiments demonstrate that Transformer-based models benefit from this richer temporal and treatment-aware representation, outperforming baselines across multiple predictive tasks.

We anticipate MIMIC-Sepsis will serve as a standardized resource for the broader research community. It offers a foundation for evaluating predictive models, exploring causal relationships, and developing reinforcement learning approaches for treatment optimization. Ultimately, we hope this benchmark facilitates reproducible, clinically meaningful advances in computational sepsis care.

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
