# OpenReview forum: "MIMIC-Sepsis: A Curated Benchmark for Modeling and Learning from Sepsis Trajectories in the ICU"
_IEEE.org/EMBS/BHI/2025/Conference — BHI 2025_

### Official Review · Reviewer_6nqJ · 2025-06-26
**A valuable and well-executed contribution that provides the research community with a much-needed reproducible, modern benchmark for sepsis modeling, highlighting the critical importance of incorporating treatment variables.**

**Confidence:** 5
**Clarity Of Writing:** excellent
**Clinical Significance:** great
**Methodological Novelty:** good
**Overall Rating:** 7
**Final Rating:** 7

**Experiments And Results:**

great

**Questions For The Authors:**

1. Implement Robust Evaluation with Cross-Validation: The evaluation should be strengthened by using k-fold cross-validation instead of a single train/test split. This will produce more stable and reliable estimates of model performance and increase confidence in the results.

2. Add Post-Hoc Statistical Testing: To support the claims of model superiority, the authors should perform post-hoc statistical tests (e.g., paired t-tests on the cross-validation fold results, or DeLong's test for comparing AUROC values). This will determine if the performance differences between models are statistically significant.

3. Sensitivity Analysis on Observation Window: To strengthen the conclusion that treatment variables are less impactful for static tasks due to the short time window, a sensitivity analysis would be beneficial. The authors could show how model performance changes if the window for the static tasks is extended to 12 or 24 hours.

3. Deeper Dive into Confounding: While a full causal analysis is likely beyond the scope, a more detailed discussion or a simple stratified analysis could add value. For instance, the authors could analyze model performance within strata of initial disease severity (e.g., based on the initial SOFA score) to see how treatment information is used differently for patients who present with similar acuity.

4. Justification for Imputation Strategy: The paper lays out a clear tiered imputation strategy. It would be improved by adding a brief justification for the specific methods chosen (e.g., "KNN was chosen for moderate missingness as it can capture complex relationships between variables, which is superior to simpler methods like mean imputation for physiological data.").

5. In Figure 3, p-values are reported as "p-value: 0.0000". The standard convention is to report this as "p < 0.0001".

6. In the Abstract and Introduction, the authors mention that the cost of sepsis is staggering. It would strengthen the impact to cite the source for the US$32,000 per patient figure.

**Strengths:**

1. Focus on Reproducibility: The primary strength is the development and commitment to releasing a transparent, well-documented data processing pipeline. This directly addresses a major crisis in computational medicine, enabling other researchers to verify, build upon, and fairly compare their models against these benchmarks.

2. Use of a Modern, Large-Scale Dataset: By building the cohort from MIMIC-IV, the authors provide a resource that is more reflective of current clinical practices, larger, and more comprehensive than the frequently used MIMIC-III dataset.

3. Systematic Inclusion of Clinical Interventions: A standout contribution is the explicit inclusion and analysis of four key sepsis interventions. The experiments clearly demonstrate that this treatment data is not just noise but contains a vital signal that significantly improves the performance of dynamic prediction models.

4. Thoughtful Benchmark Design: The selection of four clinically relevant tasks, split between static early prediction and dynamic monitoring scenarios, provides a comprehensive framework for evaluating different facets of sepsis modeling. The use of robust metrics like AUROC and AUPRC is appropriate for clinically imbalanced datasets.

**Summary Of The Paper:**

The authors introduce MIMIC-Sepsis, a new, curated cohort and benchmark framework derived from the comprehensive MIMIC-IV database. The work aims to address significant limitations in prior computational sepsis research, namely the reliance on outdated datasets, the lack of reproducible preprocessing pipelines, and the frequent omission of clinical treatment data. The authors detail a transparent pipeline for cohort creation using Sepsis-3 criteria, which includes 35,239 patients with time-aligned physiological data and harmonized interventions (vasopressors, fluids, antibiotics, mechanical ventilation). They establish four benchmark tasks: two static (in-hospital mortality, length of stay) and two dynamic (vasopressor requirement, septic shock), and evaluate several machine learning models. A key finding is that incorporating treatment data substantially improves the performance of predictive models, particularly for dynamic tasks using a Transformer architecture. The paper concludes that MIMIC-Sepsis provides a robust and extensible foundation for advancing machine learning in critical care.

**Weaknesses:**

1. Limited Generalizability: As the authors acknowledge, the dataset is derived from a single academic medical center. Clinical practices, patient populations, and documentation habits can vary significantly between institutions, which may limit the generalizability of models trained on this benchmark.

2. Unaddressed Confounding by Indication: The paper correctly identifies that the observed negative association between treatments and outcomes (Figure 3) is likely due to confounding by indication (i.e., sicker patients receive more aggressive treatments). However, the predictive models do not attempt to disentangle this. While the goal is prediction, not causal inference, this limits the interpretability of how the model uses treatment information and could be misleading if not handled with care.

3. Static Prediction Window: The choice of a 6-hour window for the static prediction tasks (mortality, LOS) is short. While this simulates a rapid assessment scenario, many crucial treatments and physiological responses may occur after this period. This limited window may be the reason for the minimal performance gain when adding treatment variables, potentially understating their importance in early risk stratification.

4. Definition of Infection: The use of antibiotic administration or positive cultures to define the time of suspected infection is a standard and reasonable approach. However, antibiotics are sometimes given prophylactically, which could introduce noise and inaccuracies in defining the true onset time of infection for some patients.

5. Evaluation Based on a Single Data Split: The evaluation relies on a single 80/20 train/test split. This approach can produce results that are not robust, as the performance metrics may be sensitive to the specific composition of that particular split. A more rigorous evaluation is needed to ensure the findings are generalizable.

6. Lack of Statistical Comparison Between Models: The results tables show that the Transformer model performs best, but there are no statistical tests to confirm that its performance is significantly better than the other models. A small difference in metrics could simply be due to random chance.

---

### Official Review · Reviewer_M3g7 · 2025-07-03
**A Reproducible Benchmark for Modeling Sepsis Trajectories**

**Confidence:** 4
**Clarity Of Writing:** great
**Clinical Significance:** great
**Methodological Novelty:** fair
**Overall Rating:** 5
**Final Rating:** 5

**Experiments And Results:**

good

**Questions For The Authors:**

How were the hyperparameters for the LSTM and Transformer models selected?

**Strengths:**

- Timely and useful contribution: Addresses the need for a reproducible benchmark based on a modern dataset (MIMIC-IV).
 - Transparent pipeline: Offers clear and modular steps for cohort selection, preprocessing, and variable inclusion.
 - Inclusion of treatment features: Captures key dynamics of ICU care and reflects real-world clinical practice better than many prior works.
 - Well-structured benchmark: Clearly defined tasks, metrics, and baseline comparisons make it accessible for the broader ML-for-health community.
 - Strong experimental results: Demonstrates the added value of intervention variables and Transformer architectures in dynamic prediction tasks.

**Summary Of The Paper:**

The paper presents a benchmark dataset derived from MIMIC-IV consisting of 35,239 ICU patients meeting Sepsis-3 criteria. It provides a preprocessing pipeline that includes time-alignment of data relative to sepsis onset, standardized units, structured imputation strategies, and harmonization of treatment interventions (e.g., vasopressors, fluids, antibiotics, ventilation). The authors define four benchmark tasks: in-hospital mortality prediction, length-of-stay regression, vasopressor requirement prediction, and septic shock prediction. These tasks are framed either as static predictions from early observations or dynamic rolling-window forecasts. Empirical evaluation using linear models, LSTMs, and Transformers shows that inclusion of treatment features improves predictive performance, particularly for dynamic tasks. The benchmark is positioned as a foundation for future work in risk stratification, causal inference, and reinforcement learning in sepsis care.

**Weaknesses:**

- Limited modeling diversity: Only linear models, LSTMs, and Transformers are included. Other strong baselines (e.g., XGBoost, Temporal Convolutional Networks) are omitted, making it hard to gauge relative task difficulty.
 - Unclear hyperparameter tuning: The paper does not describe how hyperparameters were selected or tuned for the models, which limits reproducibility and may affect performance claims.
 - Minimal statistical analysis: No confidence intervals or significance tests are provided for performance metrics across models, making it difficult to assess robustness.

---

### Official Review · Reviewer_hfwg · 2025-07-07
**MIMIC-Sepsis: A Curated Benchmark for Modeling and Learning from Sepsis Trajectories in the ICU**

**Confidence:** 5
**Clarity Of Writing:** good
**Clinical Significance:** good
**Methodological Novelty:** fair
**Overall Rating:** 6
**Final Rating:** 6

**Experiments And Results:**

good

**Questions For The Authors:**

Please expand or add supplementary material that helps alleviate the weakenesses.
Please clarify the outlier management (mentioning the clinical thresholds used).
Please clarify and justify why an interval of 4 hours was chosen for feature aggregation, which potentially limits the usefulness of the dataset.

**Strengths:**

- The pipeline is transparent and reproducible, facilitating adoption of this derived dataset
- Benchmark tasks reflect real clinical decision points (static and rolling window)
- Up-to-date cohort for sepsis related study

**Summary Of The Paper:**

The paper introduces MIMIC-Sepsis, a cohort derived from the MIMIC-IV database intended to serve as a framework for sepsis trajectory studies. This dataset includes 35,239 ICU patients, and includes data from various MIMIC-IV tables such as vasopressors, fluids, mechanical ventilation and antibiotics. The authors test benchmark tasks: mortality prediction, length-of-stay regression, vasporessor need, and shock onset classification. They describe the pre-processing pipeline which will be released after peer review.

**Weaknesses:**

The definitions of the benchmark tasks lack elaboration/clarification (e.g. positive label prevalence, censoring rules) which can help reproducibility and increase completeness of the report.
The vasopressor need task may be too easy (achieving high AUROC even with a linear model), the authors may want to highlight the dataset usefulness with a slightly harder benchmark task.
Publish data dictionary for users less familiar with MIMIC-IV and to facilitate adoption of the dataset.

---

### Official Review · Reviewer_iPSN · 2025-07-20
**MIMIC-Sepsis: A Curated Benchmark for Modeling and Learning from Sepsis Trajectories in the ICU**

**Confidence:** 3
**Clarity Of Writing:** good
**Clinical Significance:** great
**Methodological Novelty:** great
**Overall Rating:** 7

**Experiments And Results:**

good

**Questions For The Authors:**

(1) Please use the correct font size for the word 'Fig.' and 'TABLE'

(2) The font size of all figures should be enlarged. Especially (x axis, y axis label)

(3) Please include the explanation or reason of using violin plots in figure 2.

**Strengths:**

The paper content structure is good with suitable references.  The results were depicted with clear figure illustrations except the font size are small. The significance is mentioning two contributions which can reach out the study's aim and add value the current study and cutting-edge research that are using MIMIC data.

**Summary Of The Paper:**

The paper well proposes MIMIC-Sepsis, a curated cohort and benchmark framework derived from the MIMIC-IV database by standardizing dosage units, apply multi-level imputation strategies, and transform event-based clinical data into structured longitudinal format. The paper mentions the contribution with two folds. The first contribution relates with a reproducible cohort construction pipeline with harmonized clinical variables, and the second contribution relates with an evaluation framework that supports time-aware modeling of treatment dynamics and outcomes. The paper mentions the integrating temporal sequences of clinical data with treatment information improves model performance and suggests that to use MIMIC-Sepsis as a public benchmark for advancing machine learning applications in sepsis and critical care based on the study occurrence.

**Weaknesses:**

Overall, there is no the significant weakness.